# Sulfur-Modified Carbon Nanotubes for the Development of Advanced Elastomeric Materials

**DOI:** 10.3390/polym13050821

**Published:** 2021-03-07

**Authors:** Pilar Bernal-Ortega, M. Mar Bernal, Anke Blume, Antonio González-Jiménez, Pilar Posadas, Rodrigo Navarro, Juan L. Valentín

**Affiliations:** 1Instituto de Ciencia y Tecnología de Polímeros (CSIC), C/Juan de la Cierva 3, 28006 Madrid, Spain; pposadas@ictp.csic.es (P.P.); rnavarro@ictp.csic.es (R.N.); 2Department of Elastomer Technology and Engineering, University of Twente, Driener-Iolaan 5, 7522 NB Enschede, The Netherlands; a.blume@utwente.nl; 3Dipartimento di Scienza Applicata e Tecnologia, Politecnico di Torino, 15121 Alessandria, Italy; maria.bernal@polito.it; 4Materials Science and Engineering Area, Rey Juan Carlos University, C/Tulipán s/n, 28933 Móstoles, Spain; antonio.gonzalezj@urjc.es

**Keywords:** carbon nanotubes, sulfur, functionalization, natural rubber

## Abstract

The outstanding properties of carbon nanotubes (CNTs) present some limitations when introduced into rubber matrices, especially when these nano-particles are applied in high-performance tire tread compounds. Their tendency to agglomerate into bundles due to van der Waals interactions, the strong influence of CNT on the vulcanization process, and the adsorptive nature of filler–rubber interactions contribute to increase the energy dissipation phenomena on rubber–CNT compounds. Consequently, their expected performance in terms of rolling resistance is limited. To overcome these three important issues, the CNT have been surface-modified with oxygen-bearing groups and sulfur, resulting in an improvement in the key properties of these rubber compounds for their use in tire tread applications. A deep characterization of these new materials using functionalized CNT as filler was carried out by using a combination of mechanical, equilibrium swelling and low-field NMR experiments. The outcome of this research revealed that the formation of covalent bonds between the rubber matrix and the nano-particles by the introduction of sulfur at the CNT surface has positive effects on the viscoelastic behavior and the network structure of the rubber compounds, by a decrease of both the loss factor at 60 °C (rolling resistance) and the non-elastic defects, while increasing the crosslink density of the new compounds.

## 1. Introduction

Over the last few years, alternative fillers have attracted a huge interest in the development of lightweight materials [1,2]. The design and manufacturing of new materials with lower weight is of major importance in some industries such as automotive or aerospace [3]. The reduction in the fuel consumption, and therefore of the CO_2_ emissions, is the main motivation for research into these types of materials [4,5,6]. The most frequently investigated alternative fillers are clays, fibers, graphene, and carbon nanotubes [7,8]. Among them, carbon nanotubes (CNTs) stand out in the rubber field due to their great potential as a reinforcing filler. For this reason, in recent years, they have been introduced into rubber matrices for obtaining high performance materials [9,10,11,12,13,14,15,16,17,18].

The improvement of the properties of polymer/CNT composites mainly depends on the degree of dispersion of CNTs, the interactions between the filler and the matrix, and the inherent properties of the filler [1,18,19]. Due to the weak forces such as van der Waals and π–π interactions that hold CNTs together, these tubes tend to entangle with each other, forming big agglomerates known as *“bundles”*. This phenomenon is the main cause of their poor dispersion when they are used as filler in polymer matrices, reducing the expected enhancement of the properties of the final compound [1,18,19]. In this framework, in a previous study [20], it was demonstrated that these promising materials could have some limitations in applications in high-performance tire tread compounds (mainly related to the rolling resistance and fuel consumption) because of the difficulty of dispersion of CNTs in rubber matrices (high filler networking), the strong influence of CNTs on the vulcanization process (low crosslink density and high network defects), and the nature of filler–rubber interactions (high energy dissipation associated to the rupture of filler–rubber interactions at high strain amplitudes). For these reasons, the main aim of this work was to minimize these limitations, thus obtaining enhanced rubber–CNT compounds with reduced rolling resistance properties.

Although there are different methods to reach a better dispersion of CNTs in the rubber matrices [19,21,22,23,24,25,26,27,28,29,30], the surface modification of the CNT is one of the most widely-used approaches to increase the surface reactivity of these nanoparticles and improve their interaction with the polymer matrices [31,32,33,34,35,36,37,38,39]. The introduction of appropriate functional groups into the surface of the particles could lead to a better interaction of the filler with the matrix and therefore to an improvement in the final properties of the rubber nanocomposites. The most widely used functionalization method of CNTs is oxidation [19,39,40,41,42,43,44,45,46,47,48]. This method can be performed as a final treatment or as the first step of a series of chemical modifications. The main problem of this functionalization remains in the damage generated to the structure of the CNT, which depends of the type of solvents used, the treatment time, etc. [19,42,45]. Another common method reported in the rubber field for the functionalization of CNTs is the use of silane coupling agents [23,31,37,41]. Silanes are bi-functional structures that allow for the interaction with the filler as well as with the elastomeric network, acting as a compatibilizer between both phases. This technology has been used in the tire industry for many years to improve the interaction between silica and rubber [49]. Another frequently reported functionalization method of CNTs is the fluorination [50,51,52,53].

In this work, the selected approach was the chemical modification of the CNT surface with elemental sulfur. The functionalization process introduces functional groups to the CNT surface, which should enable the creation of covalent interactions with the polymer chains. The functionalization performed in this study consists of two steps: (1) oxidation of the nanoparticles in order to introduce oxygen bearing groups at the surface by using a chemical approach, using an acid treatment (named cox-CNT) and the other one was by thermal oxidation of CNTs using an oven (named tox-CNT), and (2) modification of oxidized CNTs with sulfur for obtaining the so-named chemically modified CNTs (CCNTs) and the thermal modified CNTs (TCNTs). The sulfur-functionalization was performed with the purpose to obtain covalent rubber/filler interactions as well as a more efficient use of the sulfur in the vulcanization process. This novel approach for obtaining rubber compounds was selected to overcome the reactivity problems (absorption of the sulfur on the surface of the CNTs) during the vulcanization process observed in a previous work [20], without the need to introduce new ingredients into the formulation.

Therefore, the novelty of this work is the functionalization of CNTs with elemental sulfur to obtain natural rubber (NR) nanocomposites with enhanced rolling resistance performance compared to non-modified (pristine) CNT filled compounds.

## 2. Experimental Section

### 2.1. Functionalization of Carbon Nanotubes (CNTs)

#### 2.1.1. Pristine CNT

Multi walled carbon nanotubes (CNT) NC7000^TM^ were submitted from Nanocyl^TM^ S.A (Sambreville, Belgium). According to the data given by the supplier, they had an average diameter of 9.5 nm, an average length of 1.5 µm, and a surface area between 250–300 m^2^·g^−1^ with a carbon purity of 90%.

#### 2.1.2. Chemical Oxidation

The chemical oxidation consisted in the treatment of CNTs with a 3:1 concentrated sulfuric/nitric acid mixture H_2_SO_4_ (95% of purity)/HNO_3_ (70% of purity) (both from Sigma Aldrich, Madrid, Spain) and refluxed at 70 °C for 2 h. Then, the particles were filtered through a glass filter funnel using a PTFE membrane (0.2 μm pore size, Millipore, Burlington, MA, USA) and washed with distilled water several times until neutral pH. The nanotubes were then dried at 80 °C for 24 h. After that, 150 mL of hydrogen peroxide (H_2_O_2_) (Sigma Aldrich) was added to the nanotubes and the mixture was sonicated for two hours and filtered at room temperature. Then the chemically oxidized CNTs (named cox-CNT) were dried again at 80 °C for 24 h. According to the reported literature [19,40,54,55,56], chemical oxidation using a combination of H_2_SO_4_/HNO_3_ leads to the introduction of a majority of carboxyl groups (COOH), but also hydroxyl (OH) and carbonyl (C=O) functional groups can be found. In addition, the treatment with H_2_O_2_ might result in a higher density of C=O groups [19]. A schematic representation of the followed chemical oxidation protocol is shown in Figure 1.

#### 2.1.3. Thermal Oxidation

Thermal oxidation of CNTs consists of submitting the particles to a high temperature treatment (between 200 °C–750 °C) under an oxidizing atmosphere (air, Ar, O_2,_ etc.) [44,56]. The thermal oxidation of the CNTs was performed in a tubular quartz reactor inside a heating oven. The nanoparticles were introduced in the quartz tube at 300 °C for 45 min under an oxygen atmosphere, as shown in Figure 2. After 45 min, the oven was cooled down until it reached room temperature, then the oxidized CNTs (named tox-CNTs) were removed from the quartz tube and stored in a sealed container in air.

#### 2.1.4. Functionalization with Elemental Sulfur

The second step of the CNT functionalization was the reaction of elemental sulfur into the groups inserted in the oxidation process. Therefore, both types of oxidized CNT (i.e., by chemical (cox-CNT) and thermal (tox-CNT)) treatments were subjected to the same reaction with sulfur, providing the functionalized CNTs used in this work (named as CCNTs and TCNTs, respectively). In Figure 3, a scheme of the functionalization process of the CNTs is shown.

First, oxidized CNTs and sulfur were sonicated together for 2 h using carbon disulfide, CS_2_, as a solvent, until a dispersion solution was formed. After two hours, the solvent was evaporated at reduced pressure. After that, the reaction temperature was adjusted to 155 °C by using an oil bath and heated for 2 h under vacuum to complete the reaction. Finally, the functionalized nanotubes were washed successively to eliminate all non-grafted sulfurous species. This functionalization process was adjusted from the ones found in the literature employed for the development of lithium-sulfur batteries [57,58] In Figure 4, a schematic representation of the functionalization with sulfur is shown.

### 2.2. Materials and Preparation of Rubber Compounds

The rubber used in this study was natural rubber (NR) SMR-CV60 (Rubber Research Institute of Malaysia (RRIM)). For the preparation of the rubber compounds, ZnO and stearic acid were used as activators and sulfur and N-cyclohexyl-2-benzothiazolesulfenamide (CBS) as curatives. The accelerant/sulfur ratio (1:2) was kept constant. Pristine multi walled carbon nanotubes (CNTs) NC7000^TM^ supplied by Nanocyl^TM^ S.A. (Belgium), sulfur-functionalized CNTs (named CCNTs and TCNTs according to the applied functionalization procedure, as it was explained in the previous section) and carbon black (CB) Corax N234 supplied by Orion Engineered Carbons, were selected as fillers. The filler loading in the NR compounds varied from 1 to 20 phr (1, 3, 5, 7, 10, 15, and 20 phr) in the case of CNTs, CCNTs, and TCNTs. For the reference sample filled with CB, the rubber compound was filled with 60 phr. The sulfur content for all studied samples was maintained constant to 1 phr including the samples filled with functionalized nanotubes. In those cases, the amount of free sulfur added during the mixing process was calculated considering the content of sulfur chemically bonded on the CNT surface. Therefore, it was assumed that all sulfur in the rubber compound (i.e., free sulfur and grafted sulfur) was available to react during the vulcanization process.

The NR compounds were prepared on a Gumix laboratory two-roll mill (with a cylinder diameter of 15 cm, a length of 30 cm and a friction ratio of 1:1.15) according to the formulation shown in Table 1. The rolls were kept cold during the mixing procedure by circulating cold water (20 °C). The vulcanization process was carried out in a hydraulic press at 160 °C at the optimum vulcanization time, *t*_97_, according to the rheometer curves (RPA 2000, see Section 2.3).

### 2.3. Characterization

#### 2.3.1. Structure and Morphology of CNT

Thermogravimetric analysis (TGA) was carried out on a TA discovery (TA Instruments, USA) under a N_2_ atmosphere, by equilibrating at 100 °C for 20 min and with a heating ramp of 10 °C/min up to 800 °C. Raman spectra were acquired on a RenishawinVia Reflex (Renishaw PLC, UK) microRaman spectrophotometer equipped with a cooled charge-coupled device camera at an excitation wavelength of 514.5 nm with a laser power of 10 mW (spectral resolution and integration time of 3 cm^−1^ and 10 s, respectively). The CNT were placed on a glass slide and air-dried before the measurements.

XPS studies were performed on a VG Escalab 200R spectrometrer equipped with a hemispherical electron analyzer operated on a constant pass energy mode and non-monochromatized Mg X-ray radiation (hν = 1253.36 eV) at 10 mA and 12 kV. The samples were first placed in a copper holder mounted on a sample rod in a pre-treatment chamber of the spectrometer and then degassed at room temperature for 1 h before being transferred to the analysis chamber.

The data analysis of the XPS studies was performed with the XPS 4.1 peak software. A Shirley background function was employed to adjust the background of the spectra. Atomic ratios (at%) were calculated from the experimental intensity ratios and normalized by atomic sensitivity factors (carbon 0.25, oxygen 0.66, and nitrogen 0.42). The C1s peak was fitted considering the contribution of the C–C bond sp2-like using an asymmetric peak (Doniach-Šunjić shape), previously calculated on freshly cleaved highly oriented pyrolytic graphite (HOPG) (ZYH grade, Mikromasch), obtained an asymmetry index (α) of 0.115. The curve fitting was performed using a Gaussian (80%)–Lorentzian (20%) peak shape by minimizing the total square-error fit. The full width at half-maximum (FWHM) of each peak was maintained between 1.3 and 1.4 eV. The C1s spectra was deconvoluted into several peaks: C–C sp2 with a binding energy at 284.4 ± 0.1 eV, C–C sp3 at 285.0 ± 0.1 eV, C–OH at 285.7 ± 0.1 eV, C–O–C at 286.6 ± 0.2 eV, O-C=O at 288.0 ± 0.1 eV, C=O at 289.0 ± 0.1 eV, and π–π* shake-up satellite peak from the sp2-hybridized C atoms at 291.0 ± 0.2 eV [59].

#### 2.3.2. Rubber Compounds

The vulcanization process was studied on a Rubber Process Analyzer, RPA 2000 from Alpha Technologies (Wiltshire, UK) at 160 °C by applying a deformation of 6.98% at a frequency of 1.667 Hz. The time to reach 97% conversion according to the vulcanization curve of each compound, *t*_97_, was used as molding time for obtaining the vulcanized samples in a hydraulic press.

The electrical conductivity of the rubber compounds was determined on an ALPHA high-resolution dielectric analyzer (Novocontrol Technologies GmbH, Hundsangen, Germany) over a frequency range window of 10^−1^–10^7^ Hz at room temperature. The measurements were performed on rubber samples of 2.5 × 2.5 × 0.1 mm^3^ between two-parallel gold-plated electrodes.

Field emission scanning electron microscopy (FESEM) studies were performed in order to study the dispersion of pristine and functionalized CNTs in the rubber compounds. The samples were investigated by a high-resolution field emission scanning electron microscope (ZEISS MERLIN 4248) by directly depositing the CNT on adhesive tape.

Tensile test experiments were determined using a universal mechanical tester (Instron 3366 series, Norwood, MA). Test specimens were 2 mm thick, with a test length of 20 ± 0.5 mm and a width of the narrow portion of 4 ± 0.1 mm, according to the ISO 37 (die type 2) standard. The strain speed was 200 mm/min. For each compound, five specimens were tested at room temperature.

Dynamic mechanical measurements of vulcanized samples were carried out in a TA Q8000 dynamic mechanical analyzer (TA Instruments, Inc.). Dumbbell geometry samples guided by ASTM D638 with a 1 mm thickness were used. Dynamic mechanical experiments were loaded under tension with an oscillatory deformation. Amplitude of 20 μm, frequency of 1 Hz, and “force track” (the ratio of static to dynamic forces) of 108% were applied while the temperature was changed from −80 to 100 °C, with a heating rate of 2 °C min^−1^. Strain sweep experiments were also carried out with the TA Q8000 dynamic mechanical analyzer on vulcanized samples at 40 °C by measuring the storage modulus (*G*′) and the loss modulus (*G*″) at different shear strain amplitudes (0.04 to 40%) at 10 Hz frequency.

Proton double quantum (1H DQ) NMR experiments were run using the improved pulse sequence of Baum and Pines [60,61] and were carried out at 80 °C on a Bruker minispec mq20 spectrometer operating at 0.5 T with 90° pulses of 3 µs length and a dead time of 12 µs. The temperature of the samples was controlled with a temperature controller BVT3000 through an air flow. The NMR data analysis procedure [60,62] requires the identification, quantification, and subtraction of slower relaxation signal tail from the reference intensity (*I_ref_*), which is related with non-elastic network defects [63] (e.g., dangling chain ends and loops) to obtain a sum of multiple quantum decay curve *I_ΣMQ_* that codifies information about rubber dynamics [61]. Thus, this signal is used to realize a point-by-point normalization of DW intensity, *I_DQ_*. The analysis of the so-obtained normalized DQ build-up curve (*I_nDQ_* = *I_DQ_*/*I_ΣMQ_*) by using a numerical inversion procedure based on fast Tikhonov regularization is used to obtain the actual distribution function of residual couplings [64], *D_res_*, which is directly related to the spatial distribution of rubber network constraints in the sample [60,65]. More details about the ^1^H DQ-NMR measurement and analysis procedure applied to the CNT-NR samples are given in the Supporting Information section in the reference [20].

## 3. Results

### 3.1. Characterization of Functionalized CNT

#### 3.1.1. Thermogravimetric Analysis (TGA)

TGA was performed to evaluate the amount of functional groups chemically bonded to the CNT surface. Figure 5 displays the thermogravimetric curve and the derivative thermogram (DTGA) curve of pristine CNTs, CCNTs, TCNTs, and elemental sulfur. It can be observed on the thermogram of pristine CNTs that there was almost no weight loss until approximately 600 °C because of the stability of the graphene structure. Above 600 °C, a small weight loss occurred, which was ascribed to the presence of metal impurities from CNT synthesis and amorphous carbon [66,67]. In the case of functionalized CNTs, a significant weight loss could be observed in the range of temperatures between 150 °C and 350 °C, which corresponded to the sulfur functional groups bonded to the CNT surface, as can be concluded by a comparison with the TGA curve of elemental sulfur. Nevertheless, the sulfur degradation in the TCNT sample takes place at higher temperatures than in the case of the CCNT sample. This effect may be ascribed to the different oxidation process used in the first step of the functionalization [19,36,40,44,47,54,55]. Hence, the amount and nature of oxygen-bearing groups in each treatment may have an impact on the further functionalization step with sulfur. It can also be noted that the CCNT sample (with the acid treatment) showed a continuous thermal degradation. This continuous weight loss can be associated with structural damage, which can be explained by the more aggressive treatment to which these particles have been subjected during the chemical oxidation with strong acids [67].

The different functionalization reactions carried out in this work had the corresponding sulfur/CNT weight ratio fixed by the formulation of each rubber compound. The weight loss of each CNT sample from 200 °C to 600 °C was used to determine the fraction of attached sulfur to the CNT surface, as shown in Table 2. It is worth noting that this value also includes the small weight loss associated with the oxygen-bearing groups attached during the oxidation step. Considering the fraction of grafted functional groups on the CNT surface, the amount of free sulfur that should be added for achieving 1 phr of total sulfur in each rubber compound was calculated (see Table 2). Additionally, the amount of functionalized CNT was also adjusted in order to maintain the same amount of nanotubes for a given filler volume fraction, independently of the type of filler in the samples (see Table 2). In this way, it is possible to compare the results of rubber compounds filled with pristine CNTs with those loaded with functionalized CNTs.

#### 3.1.2. Raman Spectroscopy

Raman spectroscopy was performed in the CNT samples to evaluate the presence of defects on the surface of the nanoparticles. Carbon-based materials present two characteristic bands: the *G* band assigned to the in-plane vibration of the C–C bond, at ~1570 cm^−1^, and the *D* band at ~1350 cm^−1^, attributed to the presence of disorder or defects in carbon systems [68,69,70]. The ratio between the intensity of these two bands can be used as a measurement of the amount of disorder in the surface of the nanotubes. Hence, higher values of *I_D_/I_G_* are related to the presence of a large number of defects on the nanoparticles.

The Raman spectra for pristine and modified CNTs are shown in Figure 6. In the case of modified CNTs, only a reference sample for each treatment was characterized (e.g., the samples modified with 1:1 S/CNT weight ratio). The *I_D_/I_G_* ratios for the different particles were calculated from the Lorentzian deconvolution of the *D* and *G* bands, respectively (see Table 3). The increasing *I_D_/I_G_* ratio with both surface modification treatments compared to the pristine CNTs confirms the inclusion of chemical groups on their surface. This increase was slightly higher for the CCNTs, where more oxygen groups formed at the surface of the particles because of the oxidation of CNT with strong acids compared to the thermally oxidized TCNTs.

#### 3.1.3. X-ray Photoelectron Spectroscopy (XPS)

The chemical bonding of different moieties during the CNT functionalization process were analyzed by the deconvolution of C_1s_ and S_2p_ XPS spectra (see Figure 7). The C_1s_ spectrum for pristine CNTs (Figure 7a) is dominated by a single peak at 284.6 eV, assigned to the *sp*^2^ C–C bonds of graphitic carbon. The deconvolution of the C_1s_ peak into different fitting curves allowed us to observe a peak between 285–286 eV assigned to *sp*^3^ C-atoms and peaks in the region of 285.5–290.5 eV assigned to carbon attached to different oxygen groups. It can be observed that the C_1s_ spectrum shows also a broad weak component at around 291 eV which corresponds to π-π* transition of carbon atoms in the graphene structure.

In the case of functionalized nanotubes (Figure 7b,d), reference samples for CCNTs and TCNTs with a 1:1 S/CNT weight ratio, a peak at 286 eV, corresponding to the C–S bond, have been detected. This confirms the presence of sulfur groups covalently bonded at the CNT surface. The fraction of oxygen and sulfur groups from the C_1s_ spectra were estimated by first calculating the area percent of each corresponding oxygenated and sulfidic group, and then, the atomic percentage was obtained using the O/C and S/C ratios for the individual functional groups [71].

An increase in the oxygen groups at the CNT surface of ~12% for the chemical oxidized particles and ~10% for the thermal oxidized CNT was reached. Additionally, a presence of 18% and 17% of sulfur groups was estimated for the chemical and thermal oxidized CNTs, respectively [72]. These results are in agreement with the ones obtained by TGA analysis.

Figure 7c,e displays the S_2p_ spectra for CCNT and TCNT particles, respectively. A double peak can be observed at approximately 163 and 165 eV, ascribed to the signals of elemental sulfur (S_2p_^1/2^ and S_2p_^3/2^). Moreover, in Figure 7c, at 166 eV, the single peak was assigned to the oxygenated sulfur compounds (S–O) for the chemically oxidized CNT [38,57,72,73,74]. The presence of these oxygenated sulfur groups indicates that the sulfur is covalently bonded to the CNT surface in the chemically oxidized particles.

### 3.2. Characterization of Rubber Compounds

#### 3.2.1. CNT Dispersion

The dispersion of pristine and functionalized CNTs in the rubber matrix was analyzed by SEM microscopy using the samples loaded with 10 phr of filler (see Figure 8). The images of these compounds showed that pristine CNTs (Figure 8a) and thermal oxidized CNTs (Figure 8c) were well dispersed in the NR matrix despite the high amount of filler in the compounds. However, in the case of the rubber compound filled with CCNTs, some differences could clearly be observed. As can be seen in Figure 8b, nanotubes chemically modified were more entangled, forming bigger agglomerates that could impact on the properties of the filled compounds compared to pristine CNTs such as reinforcement or dynamical mechanical properties, as will be discussed in the following sections.

#### 3.2.2. Vulcanization Process

The different cure reactions for natural rubber compounds filled with pristine and functionalized CNTs are shown in Figure 9. In all cases, the increasing fraction of nano-particles led to a decrease in the induction (*t*_0_) and vulcanization time (*t*_97_) (see Table 4), a rise in the maximum torque (*S’_max_*), and the appearance of a slight reversion process (slow decrease of *S’* after reaching its maximum value).

By comparing the differences between fillers, it was observed that the rise in *S’*_max_ (mostly associated with the reinforcing effect of fillers and rubber crosslink density) [75,76] was similar for CNTs and TCNTs, but was lower in the samples filled with CCNTs. The possible explanation for the low *S’*_max_ of the NR/CCNT samples is twofold: (i) The agglomeration of CCNTs and their poor dispersion in the rubber matrix, as observed by SEM images, leading to a lower reinforcing effect; and (ii) the deactivation of CBS due to the acidic moieties grafted at the CNT surface (e.g., COOH) could adsorb this accelerator, leading to a less efficient vulcanization process for the CCNT samples.

Finally, the addition of fillers seemed to promote an important increase in the minimum torque, *S’*_min_, for all the studied samples, as shown in Figure 10. This effect was higher for the samples filled with pristine CNTs, which is related to an increase in viscosity due to the formation of strong filler network structures. This result indicates that CNT functionalization may reduce the van der Waals interactions between nanoparticles, minimizing their effect on *S’*_min_.

#### 3.2.3. Electrical Conductivity

The addition of electrical conductive particles into rubber matrices led to the formation of a filler network. The concentration of filler at which this filler networking is achieved is called the percolation threshold [12,77,78]. The variation of the electrical conductivity for the studied samples is shown in Figure 11. Pristine CNTs and TCNTs present a similar behavior of a fast increase in the electrical conductivity until a plateau at 3% *v/v* (filler volume fraction) when the percolation threshold is reached.

The variation of electrical conductivity of rubber compounds with the filler fraction is more gradual for CCNTs without achieving a clear plateau, although this system reached approximately the same electrical conductivity than the others at a filler loading of 9% *v/v*. The different behavior showed by NR/CCNT compounds can be explained by the reduced aspect ratio of the chemically oxidized CNTs and their agglomeration in the rubber matrix, which decreased the probability of particle–particle interactions [79,80], resulting in a higher number of particles required to form a filler network. In addition, the surface modification of CCNTs disturbed the π-electron system of these particles, reducing the electrical conductivity properties of pristine CNTs [34,56].

#### 3.2.4. Mechanical Properties

For many applications, rubber is reinforced with different types of fillers (e.g., carbon black, silica, clays) to obtain improvements in the mechanical properties of the final products (increase in the modulus, fatigue life, abrasion, tear strength, etc.) [81,82,83,84,85,86]. The reinforcing effect of fillers in rubber matrices is the result of several molecular mechanisms involved in this complex phenomenon [87,88,89]: the rubber network, the hydrodynamic effect, the filler–filler interactions, and the filler–rubber interactions. To estimate the reinforcement effect of pristine and functionalized CNTs and their influence on the viscoelastic properties of NR, both tensile test and Payne effect (i.e., variation of the modulus through strain sweep experiments) for rubber compounds with increasing fraction of CNTs and f-CNTs were evaluated.

The reinforcement degree (R_1_), calculated as the ratio between the modulus 100%, M_100%_ of the filled and that of the unfilled samples, was studied in the non-linear regime. As can be noticed in Figure 12, the reinforcement degree showed a sharp increase with the addition of CNT at the highest volume fraction, reaching values of ca. 1300% higher than the unfilled sample in the case of pristine CNTs, 900% for TCNTs, and 700% for CCNTs. As predicted from the results obtained by rheometer curves, a decrease in the reinforcement of CNTs could be observed in the samples loaded with functionalized particles in both CCNTs and TCNTs. However, for TCNTs, the reinforcement degree was comparable with those obtained with pristine CNTs up to 4.6% *v/v*.

In addition, it can be observed that the samples filled with 4.6% *v/v* (10 phr) of pristine and thermal modified CNTs reached the same reinforcement than the reference sample with 24.6% *v/v* (60 phr) of carbon black. For CCNTs, this degree of reinforcement was achieved with 6.7% *v/v*. Therefore, all three systems reached the values of reinforcement of a traditional filler with lower filler volume fraction, almost 20% less.

The variation of the storage modulus, *G*′, was evaluated by strain sweep experiments. As shown in Figure 13, the storage modulus of filled rubber compounds showed a decrease in the applied deformation (this is named the “*Payne effect*”), while it remained practically constant in the unfilled composites. This non-linear behavior is amplified with an increasing filler volume fraction, being interpreted as the rupture of the filler network when rubber samples are subjected to high deformations and the subsequent liberation of trapped rubber in filler aggregates as well as rubber–filler bonding and debonding mechanisms [64,65,90,91,92,93,94,95]. Following this concept, the significant reduction of the Payne effect for NR/TCNT and NR/CCNT samples compared to the non-modified CNT filled samples should be related to important differences in the filler networking and/or the rubber fraction affected by the existence of filler aggregates.

Additionally, it has to be noted that for the rubber compounds filled with pristine CNTs and TCNTs, the storage modulus already presented an observable decay for samples with lower filler loading, indicating a low percolation threshold. In contrast to this, samples filled with CCNT showed this behavior only at higher filler loadings. These results align with the evidence obtained from the electrical conductivity experiments for the percolation threshold and formation of filler networking.

In order to estimate the degree of agglomeration of the CNTs and f-CNTs inside the rubber matrix, the effective shape factor, *f*_eff_ (Figure 14) was calculated according to the reinforcing effect at the maximum strain amplitude (i.e., the ratio between the storage modulus of filled compounds and unfilled samples $\frac{{G^{\prime }}_{\infty ,f}}{{G^{\prime }}_{\infty ,u}}$), following the procedure described in a previous work [20] and using Equation (1).
(1)$$\frac{{G^{\prime }}_{\infty ,f}}{{G^{\prime }}_{\infty ,u}}=\left(1+0.67f\phi_{f}+1.62f_{}^{2}\phi_{f}^{2}\right)$$

The so-obtained approaches of the shape factor were far from the theoretical value of 158 estimated according to the theoretical dimensions of CNTs (1500 nm in length and 9.5 nm of diameter) given by the supplier. This effect was especially evident above the percolation threshold, reaching values in the range of 20–30 for the studied samples with the highest filler volume fraction, independently if the carbon nanotubes were pristine or modified with sulfur. This seems to indicate that the agglomeration into bundles with the filler content of both CNTs and f-CNTs in the rubber matrix causes a critical reduction of the effective shape factor, minimizing their reinforcing properties and hindering the reach of the outstanding effects that would be expected according to their theoretical values (see Figure 14).

Only at an extremely low CNT content (below 0.5% in volume) was it possible to obtain a good dispersion of carbon nanotubes in the rubber matrix by using traditional compounding processes [20], revealing that agglomeration in the samples with surface modified nanotubes seemed to be slightly superior than in rubber compounds filled with pristine CNTs. However, these larger agglomerates of f-CNTs could not be easily broken during the strain sweep of these rubber materials, as demonstrated by the decrease in the Payne effect shown in Figure 13. The combination of these results seems to indicate that during the surface modification reaction of CNTs, these nanoparticles may tend to agglomerate and also create strong chemical bonds between them. As a consequence, the primary particles of f-CNTs that are added to the rubber matrix during the compounding step may be in fact composed of a group of inseparable nanotubes that have been chemically linked during the modification reaction, increasing the size and reducing the effective shape factor of the pristine CNTs.

The quantitative interpretation about the different aggregation state and the aspect ratio of f-CNT primary particles in comparison with pristine CNTs is limited because there are several factors that affect the estimated *f*_eff_ (e.g., the hydrodynamic effect, the immobilized polymer fraction at the CNT surface, and the occluded rubber in the structure of filler aggregates). Nevertheless, the qualitative comparison between them allows us to explain both the Payne effect as well as the slight differences in the electrical conductivity between rubber compounds filled with CNTs and TCNTs (see Figure 11), even when in both cases, the dispersion of nanoparticles into the rubber matrix seems to be quite similar according to the SEM images (see Figure 8). On the other hand, the properties shown by the NR compounds filled with CCNTs seemed to be strongly compromised (dominated) by the poorer filler dispersion observed by SEM.

This statement was reinforced by the analysis of the strain dependence of the loss modulus (*G*″) (Figure 15), which makes it possible to identify and quantify the different contributions for the energy dissipation phenomena under deformation for the studied samples [96]. The maximum peak observed in *G*″ at intermediate strains (i.e., $G_{F}^{\prime \prime }$) was related with the heat released when the rupture of the filler network is produced due to the breaking of bonds between nanoparticles. In all samples, the peak shifted toward lower strains and the $G_{F}^{\prime \prime }$ value increased with the filler fraction, as shown in Figure 16a.

Aligned with these results, the difference between the loss moduli measured at the minimum and maximum strain ($G_{D}^{\prime \prime }=G_{0}^{\prime \prime }-G_{\infty }^{\prime \prime })$, which is related with the dissipation of energy associated with the filler–rubber–filler interactions, rises with the filler volume fraction (see Figure 16b). At the minimum strain amplitude, $G_{0}^{\prime \prime }$ should be related to the energy dissipation caused by the local friction of the rubber fraction between filler particles. Consequently, the rupture of filler interactions with the strain sweep reduces this phenomenon and releases the rubber fraction trapped at the filler aggregates, reducing the loss modulus with the applied strain until it reaches the minimum value at $G_{\infty }^{\prime \prime }$ when the maximum strain is applied.

According to these statements, the different dissipation phenomena that are revealed by $G_{F}^{\prime \prime }$ and $G_{D}^{\prime \prime }$ has the same origin (i.e., the networking of primary filler nanoparticles, i.e., aggregates and agglomerates). For that reason, the trend of both moduli with the filler volume fraction for CNTs and f-CNTs were quite similar (see Figure 16a,b), showing a significant upturn in their value above the percolation threshold. Nevertheless, it is important to note that the surface modification of these nanoparticles seems to significantly reduce the energy dissipation phenomena attributed to the breakdown of filler interactions despite their larger filler aggregates. This apparent inconsistency may be explained by the formation of covalent bonds between CNT particles during the modification reaction that increases the size of the primary particles (unbreakable aggregates of CNTs).

Figure 16c shows the variation of $G_{\infty }^{\prime \prime }$ with the filler volume. The loss modulus at the maximum strain reveals the energy dissipation caused by other factors different to the filler network. For this reason, its trend was quite different to those observed for $G_{F}^{\prime \prime }$ and $G_{D}^{\prime \prime }$, respectively (see Figure 16). Nevertheless, $G_{\infty }^{\prime \prime }$ increased with the CNT content because of energy loss associated with the bonding–debonding processes of rubber segments at the filler surface, the friction of non-broken filler–rubber–filler interactions and other phenomena caused by the existence of dangling chain ends and chain loops. These non-elastic network defects are directly affected by the presence of CNTs and f-CNTs as was studied by DQ-NMR.

It has been widely reported that the presence of filler particles significantly alters the overall dynamic response of the rubber matrix [90]. The strength of the filler–filler or filler–rubber interactions as well as the dispersion of the filler affects the dynamic behavior of the rubber compounds. Rubber reinforcement using fillers is intimately related to the energy dissipations produced during the deformation of the material. These energy dissipations are of great importance in many applications, especially in the tire industry, where these dissipations affect the service performance of the final product in terms of heat generation, rolling resistance, traction, and skid. The study of these fundamental characteristics for tire materials is possible by analyzing the viscoelastic and mechanical properties of these rubber compounds [90,97,98,99].

A detailed description of the viscoelastic behavior of rubber compounds was obtained by the study of the variation of the loss factor, *tan δ,* with temperature. Based on the experience of tire producers, the value of *tan δ* correlates well with the tire grip and rolling resistance, as a function of the temperature range. The loss factor measured at a temperature in the range of 60–80 °C, most often the value at 60 °C, in conditions of constant deformation (1%) and high frequencies (10–100 Hz), similar to the ones experimented by a tire tread in its service life, is used as a prediction of the rolling resistance. A decrease in the loss factor in this temperature region implies a lower rolling resistance of the tire, and as a consequence, a reduction in the fuel consumption [90,97,98,99]. Figure 17 shows the effect of CNTs and f-CNTs in *tan δ* at 60 °C. Samples with functionalized CNTs led to a significant decrease in this factor in comparison with pristine CNTs and carbon black, overcoming some of the limitations defined in a previous work for CNTs to be used in tire tread compounds [20]. For example, the rubber compounds with 4.6% *v/v* of modified CNTs showed *tan δ* values around 35% lower than the counterpart filled with pristine CNTs. Additionally, it is important to note that the addition of up to 2.3% *v/v* of functionalized CNTs does not modify the optimum *tan δ* value obtained for the unfilled NR matrix. According to the obtained results, the sulfur modification of CNTs leads to a significant enhancement in the viscoelastic performance of NR compounds to be used in tire treads, that is, the reduction in the energy dissipation at the rolling resistance conditions because of different key effects. The first effect is related to the rubber bonding–debonding from the filler surface, which is strongly limited or suppressed. The interactions between the rubber with CNTs and CB are mainly adsorptive, allowing the energy loss in the form of heat during the dynamic deformation of the rubber compounds. However, the insertion of sulfur at the CNT surface enables the formation of covalent bonds between the rubber chains and the surface of the nanoparticles in a similar way as the modified silica technology with the use of bi-functional silanes. The second effect is ascribed to the reduction of the filler network. The breakdown of filler–filler interactions and the friction caused at the filler–rubber–filler interface is because of the existence of occluded rubber trapped in filler aggregates and agglomerates are strongly minimized because of the functionalization of CNTs. The last effect is due to the dissipative phenomena related to the rubber network structure. In the last years, it has been proven that rubber networks contain non-elastic network defects (e.g., loops and dangling chain ends) that have to be additionally considered to fully understand the elastic response of these materials [100,101]. The introduction of sulfur at the surface of the CNT led to a more efficient vulcanization process and the formation of a more homogeneous rubber network with less non-elastic defects.

The effect of the surface functionalization of CNTs with sulfur and the reactivity of these particles to generate a rubber network structure and rubber–filler interactions during the vulcanization process will be proven and evaluated in the next section of this paper.

#### 3.2.5. Rubber–CNT Network Structure

The use of time-domain NMR experiments for the characterization of rubber structure is a useful tool to obtain a deeper knowledge of the structure–property relationships for the NR compounds filled with pristine and modified CNTs at shorter time and length scales than the ones analyzed by rheological or mechanical data. The versatility of this experimental approach makes it possible to obtain both structural and dynamic information of rubber systems through the applications of advanced NMR pulse sequences [60,62,95,102]. In this work, ^1^H DQ-NMR experiments were performed in a low field spectrometer to obtain detailed information about the structure of the NR compounds and how the functionalization of CNT affects the rubber network structure. Figure 18 shows the obtained distributions of the residual dipolar couplings, *D_res_*, which is proportional to the molecular weight distributions between constraints (i.e., entanglements, cross-links and filler–rubber interactions) [60,65,95,103].

The addition of CNTs into the matrix affects the network structure in a very significant way. As shown in Figure 18, the incorporation of these particles causes the broadening of the residual dipolar coupling distributions. This behavior is visible in the three rubber systems (i.e., filled with pristine and both functionalized CNTs, respectively). The broadening of the *D_res_* distributions is caused by two different processes: the displacement of the maximum value of the distribution to lower *D_res_* values and the formation of tails in areas of high values of *D_res_*. The amplitude of these distributions provides the fraction of protons with a given dipolar coupling and the area under the curve characterization to all dipolar coupled protons in the sample (normalized to the unit). As a consequence, as already discussed in a previous work [20], the formation of tails at higher *D_res_* values with the volume fraction of CNT is related to the formation of strong interactions between the rubber and the CNT. It has been reported [104,105] that the rubber fractions immobilized by filler particles present higher values of residual dipolar couplings compared to the bulk, leading to some contrast between them and the formation of these tails, which increases the distribution width.

On the other hand, the shifting of the maximum of these distributions toward lower values of *D_res_* should be related to the influence of carbon nanotubes in the vulcanization process. This process is being hindered due to the absorption of some vulcanizing agents at the surface of the CNT. As a consequence, the average value of crosslink density of the NR matrix (identified as the *D_res_* value at the maximum amplitude) is reduced with the addition of particles, as observed more clearly in Figure 19. The addition of particles caused almost a linear decrease in the value of *D_res_*, and therefore in the crosslink density of the bulk.

Even if this characteristic behavior is observed in all three systems, it is worth noting that it was clearly reduced in the samples filled with functionalized CNTs, reaching similar values to the reference sample filled with carbon black. The introduction of sulfur in the functionalized nanotubes and the consequent formation of covalent bonds led to a different interaction between these particles and the rubber compared with the pristine CNTs and a more efficient use of sulfur during the vulcanization process. According to these results, it is possible to suggest that surface modification of CNT reduces the absorption of different ingredients of the vulcanization process in the CNT particles (especially CBS). However, it is important to note that the rubber compounds with the highest f-CNT volume fraction did not contain additional sulfur during the mixing process. This demonstrates the efficiency of the sulfur bonded to the CNT surface as an active element to react with the elastomeric chains during the vulcanization process. Furthermore, it generates a crosslinked network in a similar way as the one generated when the sulfur is added during the mixing process (see vulcanization curves, Figure 9). This means that the complex reaction between activators, accelerants, and sulfur occurs at the surface of the CNT, being possible after the reaction between the sulfurous species generated and the polymeric chains to create covalent unions. These results demonstrate that functionalized filler particles act as macro- and poly-functional crosslinks that can be combined with the tetra-functional sulfur crosslinks formed between rubber chains by the free sulfur (additionally added during the rubber compounding) to create a complex network structure.

As a consequence of the f-CNT surface reactivity and the dispersion of the filler particles, the crosslink density for samples filled with TCNTs was not reduced compared to the unfilled sample until filler volume fractions higher than 0.05 (see Figure 19). The more efficient use of sulfur during the vulcanization process with the use of functionalized nanotubes was also corroborated by the decrease in the non-coupled network defects (i.e., sol fraction, dangling chain ends and chain loops), in comparison with pristine CNTs (Figure 20). It was shown that pristine CNTs reached a maximum value of 45% of elastically inactive network defects with the addition of the highest volume fraction of filler. The functionalization of the particles led to a decrease in network defects compared to the pristine CNTs up to approximately 20% as the maximum values. However, the fraction of network defects obtained for rubber compounds filled with functionalized particles were still much higher than the ones of the sample filled with carbon black, which presented almost no variation in the number of defects compared to the unfilled counterpart. It is important to note that these defects act as energy dissipation elements and negatively affect the viscoelastic properties of the compounds.

Several investigations [20,106] have reported the non-reliability of swelling experiments for the characterization of the crosslink density of filled compounds. However, the combination of this technique with low field NMR can be used to study the degree of rubber–filler interactions and the swelling restrictions caused by the addition of filler particles. In this framework, the filler–rubber interactions of the studied compounds were analyzed by the determination of the swelling restriction ratio (*SRR*) (Figure 21). The calculation of this parameter was performed as explained in previous works and according to Equation (2) [20,106].
(2)$$\mathrm{Swelling}\;\mathrm{Restriction}\;\mathrm{Ratio}=\frac{1/M_{c,\mathrm{filled}}^{\mathrm{sw}}-1/M_{c,\mathrm{unfilled}}^{\mathrm{sw}}}{1/M_{c,\mathrm{unfilled}}^{\mathrm{sw}}}$$

As observed in Figure 21, the swelling restrictions increased with the increasing filler volume fraction in all systems. As mentioned before, these restrictions are caused by the high density of interactions between the elastomeric segments at the surface of the nanotubes, having a positive influence on the reinforcing effect of these particles. The samples filled with CCNTs and TCNTs showed lower filler–rubber interactions than pristine CNTs, in agreement with the results observed in the *D_res_* distributions (lower formation of tails in the areas of high *D_res_*), although the nature of these joints is quite different. Whereas in the case of CNTs, the rubber interacts with the particle surface via adsorptive forces, in the case of f-CNTs, the rubber segments are covalently linked to the filler surface.

The agglomeration of the particles during functionalization causes a decrease in the available surface area that can interact with the polymer chains. These can be observed by the determination of the normalized *SRR* (see Figure 21b). These normalized *SRR* were obtained using the surface area of the fillers, 250 m^2^·g^−1^ for CNTs and 118 m^2^·g^−1^ for CB. According to these results, the degree of agglomeration of the carbon nanotubes was higher with increasing filler content; for this reason, although the total number of rubber–filler interactions increased (see Figure 21a), the density of junctions per filler surface decreased. This behavior was more pronounced in the samples with functionalized CNTs, confirming that there were more aggregates than with pristine CNTs. Taking again into consideration the results from the Payne effect, these higher aggregates cannot be broken during the strain sweep experiments, reflecting that they are strongly bonded (maybe with covalent bonds) during the modification reaction.

Compared to the sample filled with 24.6% *v*/*v*, which corresponded to 60 phr of CB, the three CNT systems reached or even exceeded (NR-CNT and NR-TCNT samples) the rubber fraction joined at the filler surface of this reference sample with a much lower filler volume content of particles. The higher degree of filler–rubber interactions in all CNT systems compared to CB can be explained by the high aspect ratio of CNTs, presenting a higher effective surface area that can interact with the rubber.

## 4. Conclusions

The promising mechanical and viscoelastic properties of CNT composites are strongly compromised or limited by the difficulty of dispersing these particles through their strong influence on the vulcanization process and the adsorptive nature of rubber–filler interactions, having a detrimental effect on the final properties of the rubber compounds. The aim of this study was to obtain a better interaction of CNTs with the rubber matrix by the formation of covalent bonds through the surface modification of these particles. The sulfur modification of CNTs is a promising functionalization method that can be scalable to an industrial process. It was observed that the use of sulfur based CNTs is a suitable approach that partially overcomes two of the problems identified in the rubber compounds filled with pristine CNT, which may limit their use in high performance tire tread compounds. The use of functionalized CNTs with sulfur reduces the adsorption of accelerants and promotes the formation of covalent bonds between the rubber chains and the CNT surface during the vulcanization process. In this way, these modified nanoparticles not only act like fillers, but also as macro-crosslinks that are able to create a rubber network with higher crosslink density and lower fraction of non-elastic network defects compared with the rubber compounds filled with pristine CNTs. As a consequence, these functionalized CNTs led to an improvement in the viscoelastic properties of rubber compounds by reducing their energy dissipation and promoting a substantial decrease in the loss factor at 60 °C, which is a key parameter for the rolling resistance performance of tire tread rubber compounds.

On the other hand, functionalization with sulfur was not able to solve the trend of pristine CNTs to agglomerate into bundles, causing a critical reduction of the effective shape factor with the filler volume fraction, which has consequences on their reinforcing properties. Nevertheless, the modified nanoparticles seemed to form covalent bonds between them, increasing the resistance of CNT aggregates to be broken during rubber deformation and reducing the energy dissipation phenomena attributed to the breakdown of filler interactions.

Analyzing the two oxidation methodologies applied in this work to modify CNTs, the thermal treatment seems to be the more promising approach, although further investigations are required to improve the quantitative characterization of filler–rubber interactions for TCNT-NR that could provide a better understanding of the applicability of this promising rubber nanocomposite in tire tread compounds.

## Figures and Tables

**Figure 1 polymers-13-00821-f001:**
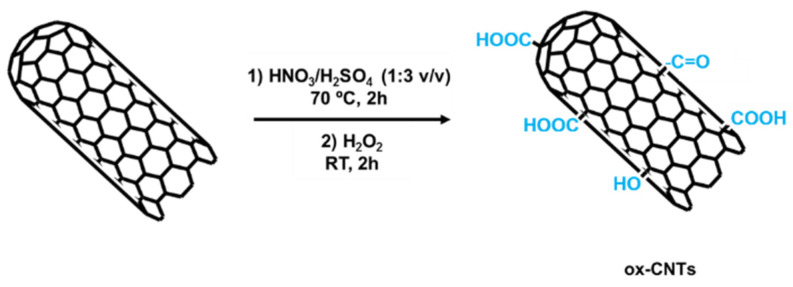
Schematic representation of the chemical oxidation of carbon nanotubes (CNTs).

**Figure 2 polymers-13-00821-f002:**
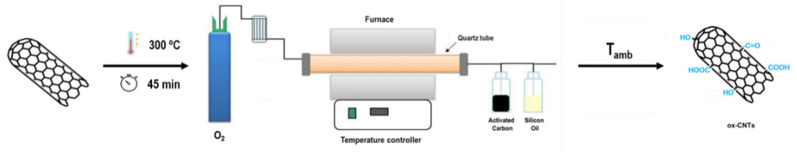
Schematic representation of the thermal oxidation of CNTs.

**Figure 3 polymers-13-00821-f003:**
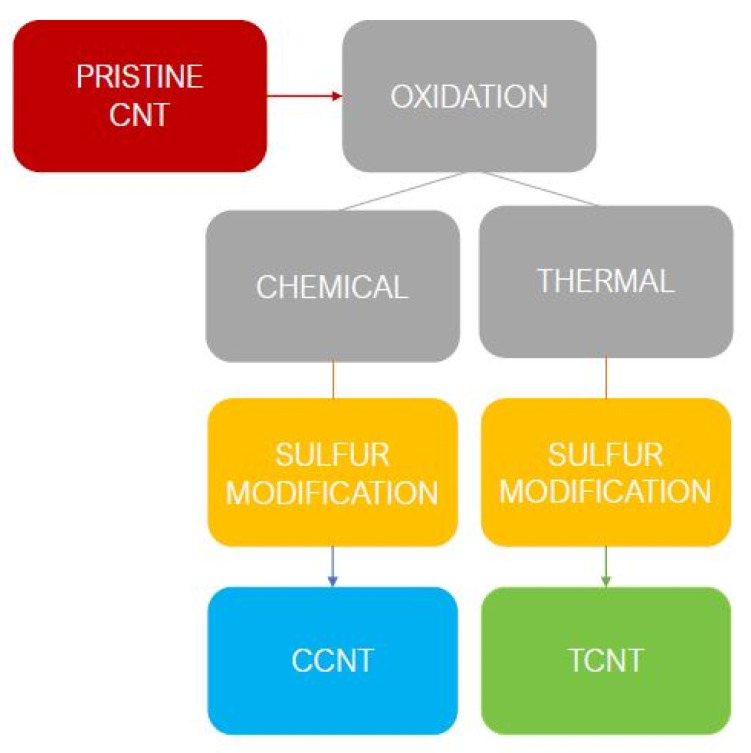
Scheme of the functionalization process of CNTs.

**Figure 4 polymers-13-00821-f004:**
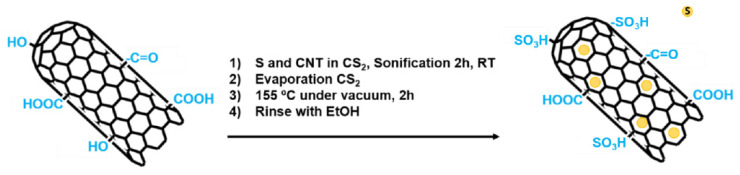
Schematic representation of the CNT functionalization with elemental sulfur.

**Figure 5 polymers-13-00821-f005:**
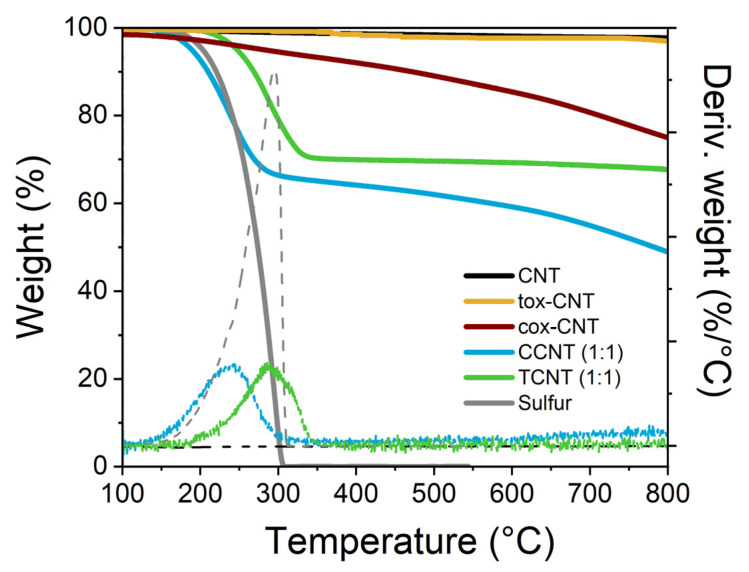
Thermogravimetric analysis (TGA) and derivative thermogravimetric analysis (DTGA) curves of pristine CNTs, chemically oxidized CNTs (cox-CNTs), thermal oxidized CNTs (tox-CNTs), reference sample of CCNT, reference sample of TCNT, and elemental sulfur. The reference samples of functionalized CNTs (independently of the applied functionalization process) correspond to the reactions in which the S/CNT ratio was 1:1 in weight.

**Figure 6 polymers-13-00821-f006:**
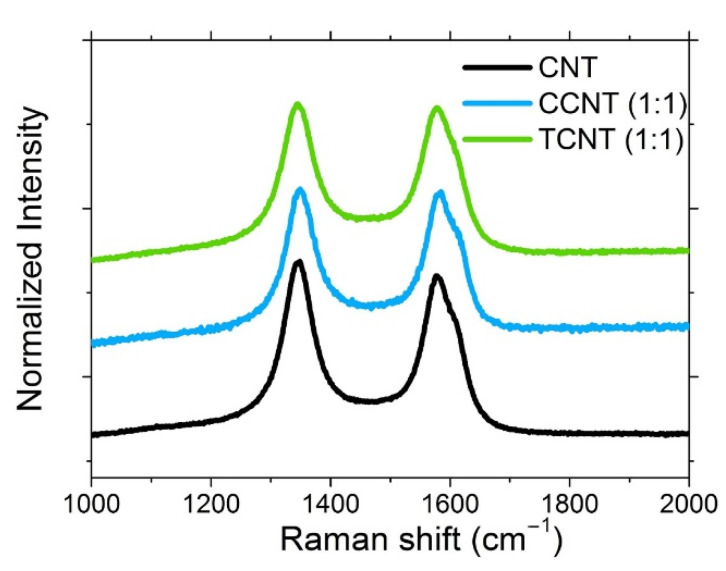
Raman spectra of CNTs, CCNTs, and TCNTs.

**Figure 7 polymers-13-00821-f007:**
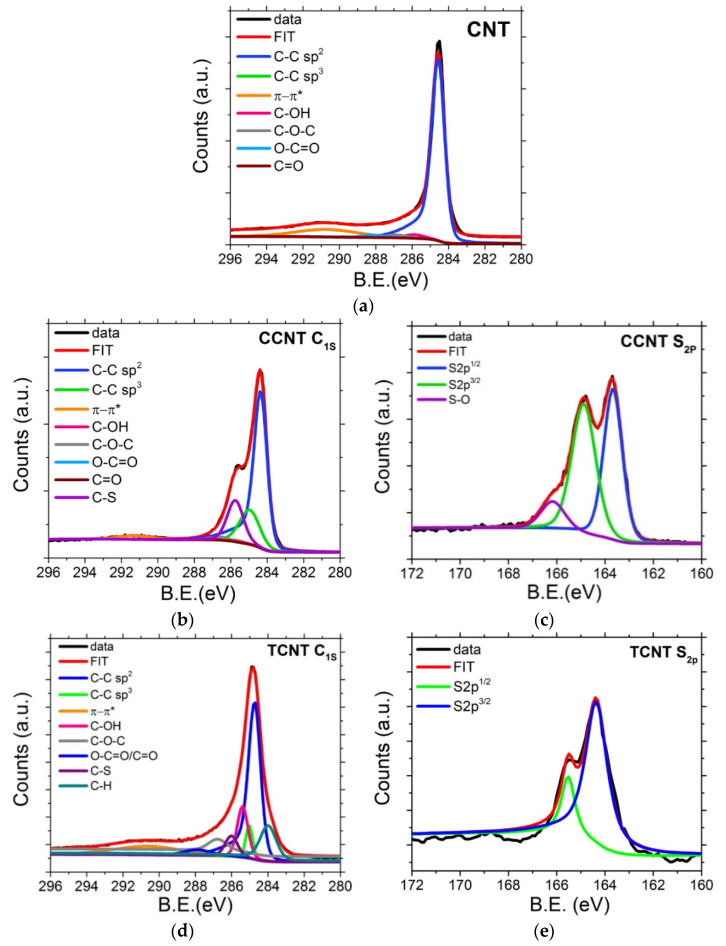
High resolution C1s and S2p X-ray photoelectron spectroscopy (XPS) spectra of (**a**) CNTs, (**b**,**c**) CCNTs, and (**d**,**e**) TCNTs. Solid lines are fitting curves of the spectra.

**Figure 8 polymers-13-00821-f008:**
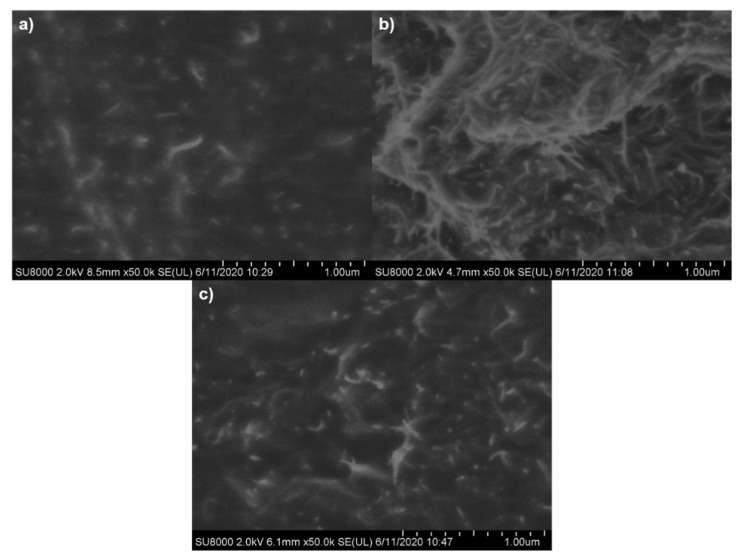
Scanning electron microscopy (SEM) images of the compounds (**a**) NR-10CNT, (**b**) NR-10CCNT, and (**c**) NR-10TCNT.

**Figure 9 polymers-13-00821-f009:**
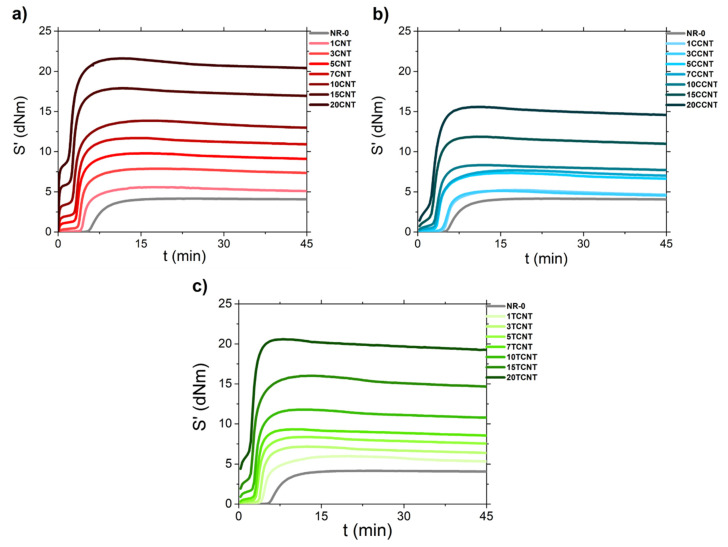
Rheometer curves of (**a**) NR/CNTs, (**b**) NR/CCNTs, and (**c**) NR/TCNTs composites.

**Figure 10 polymers-13-00821-f010:**
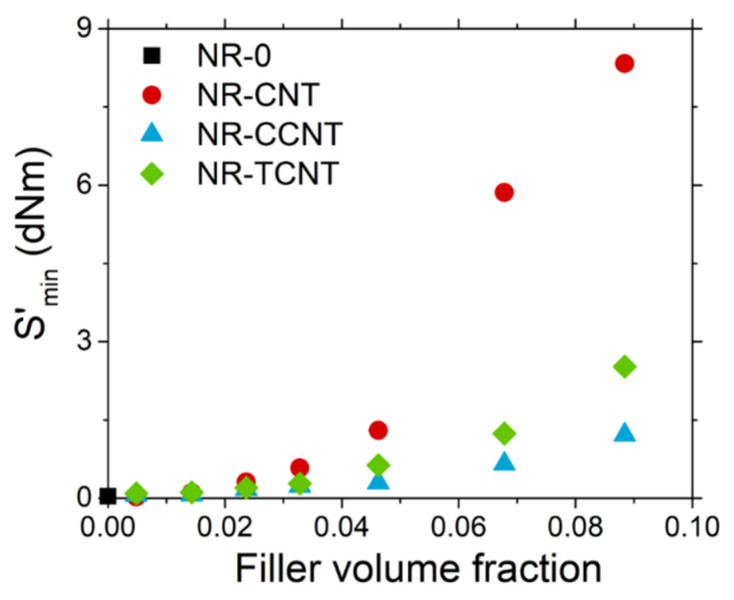
Variation of the minimum torque, *S*’_min_ with the filler volume fraction for NR/CNTs, NR/CCNTs, and NR/TCNTs.

**Figure 11 polymers-13-00821-f011:**
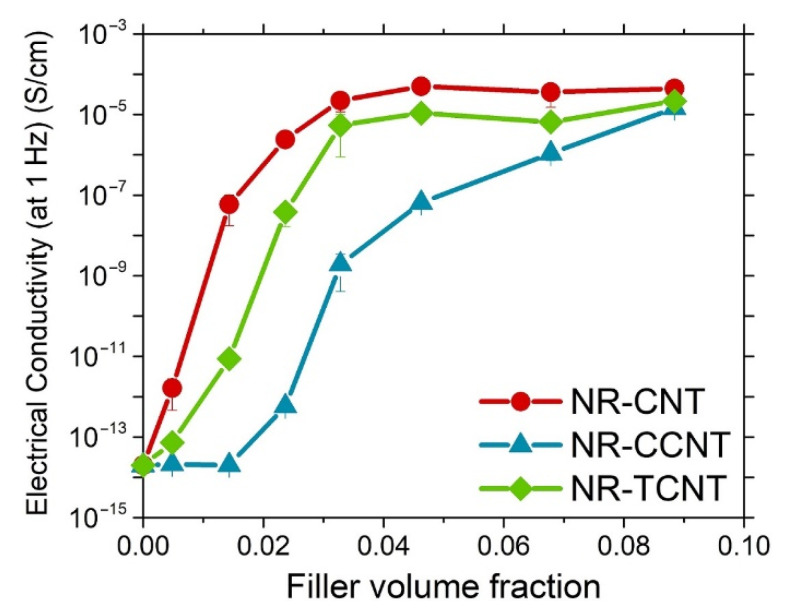
Variation of the electrical conductivity with the CNT content in natural rubber (NR) compounds.

**Figure 12 polymers-13-00821-f012:**
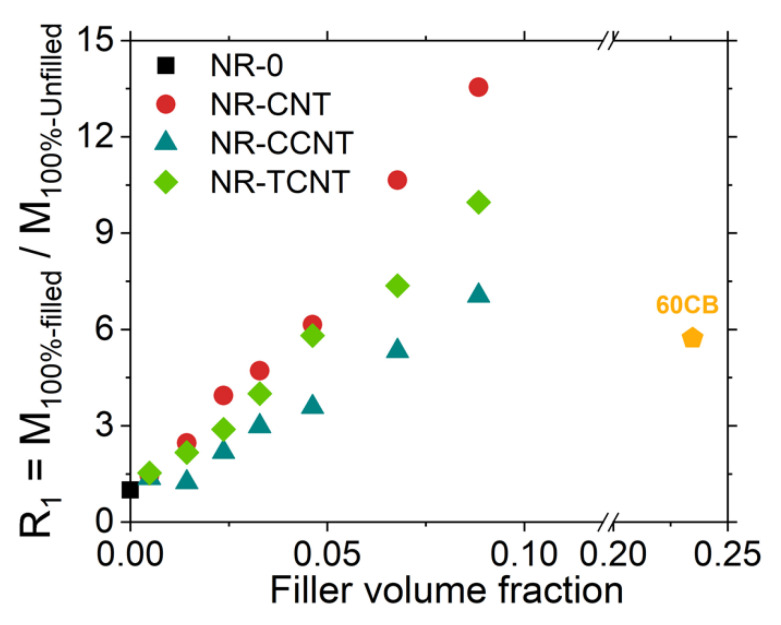
Variation of the reinforcement degree with the volume fraction of pristine (CNTs) and functionalized (CCNTs and TCNTs) carbon nanotubes compared to the performance of a sample filled with carbon black (CB).

**Figure 13 polymers-13-00821-f013:**
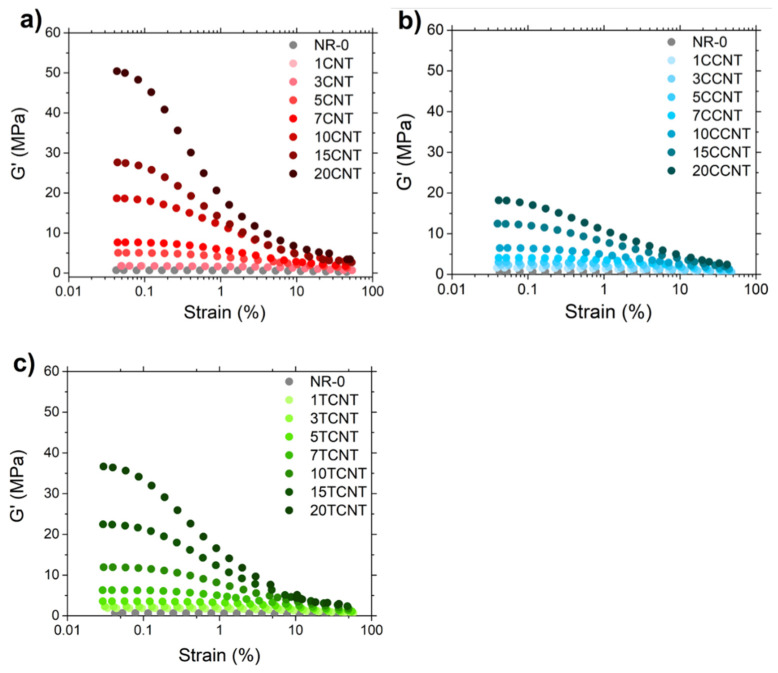
Strain dependence of *G*′ for the CNT filled compounds (**a**–**c**).

**Figure 14 polymers-13-00821-f014:**
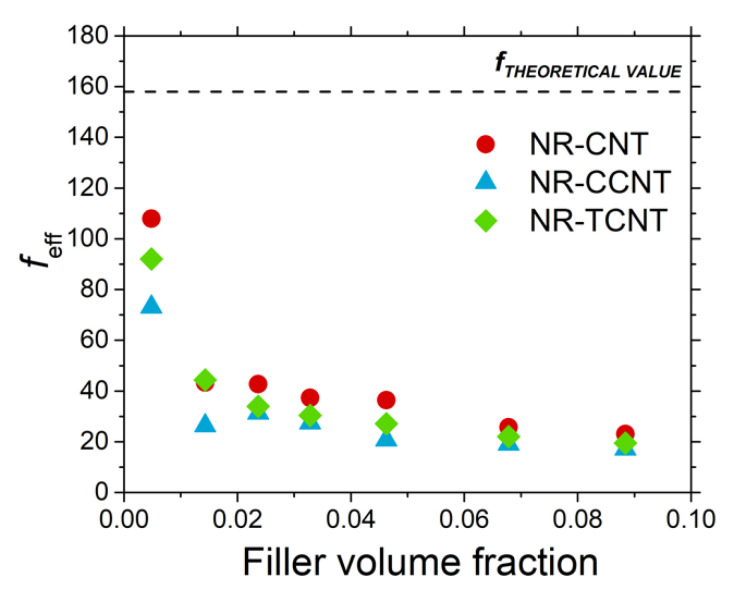
Effective shape factor of the studied rubber compounds.

**Figure 15 polymers-13-00821-f015:**
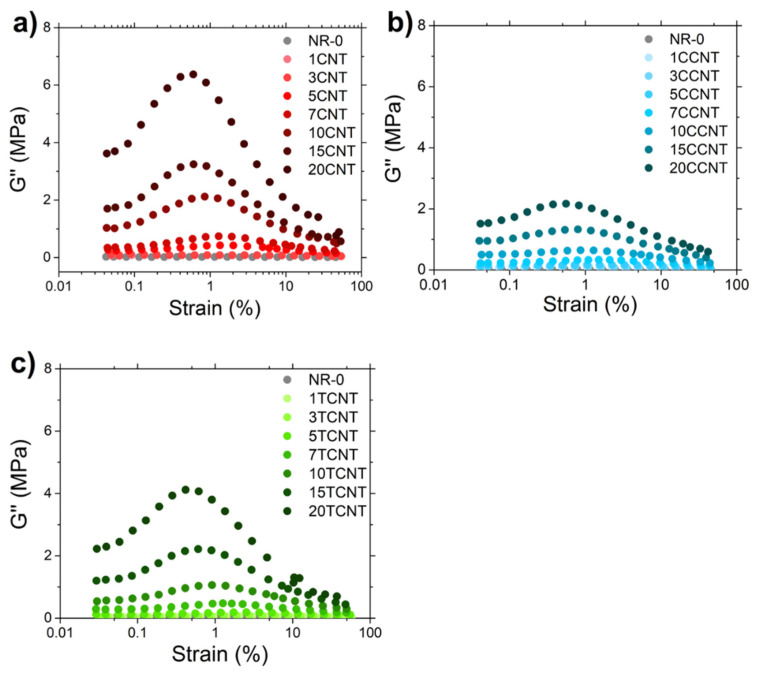
Strain dependence of *G*″ for the CNT filled samples (**a**–**c**).

**Figure 16 polymers-13-00821-f016:**
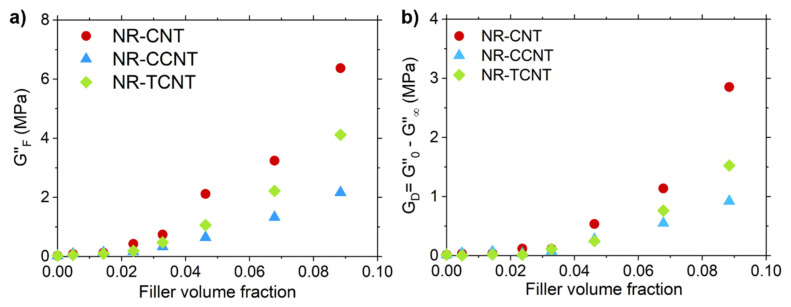
Variation of $G_{D}^{\prime \prime }$, $G_{F}^{\prime \prime }$, and $G_{\infty }^{\prime \prime }$ with the increasing filler for the studied compounds (**a**–**c**).

**Figure 17 polymers-13-00821-f017:**
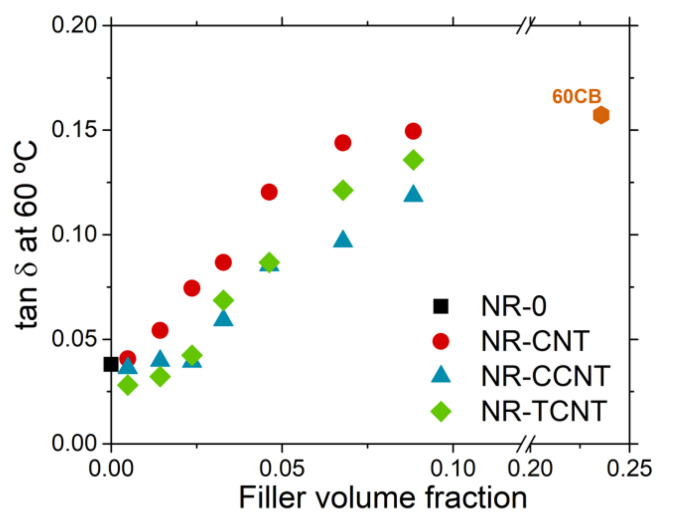
Value of *tan δ* at 60 °C for NR/CNT, NR/CCNT, and NR/TCNT composites.

**Figure 18 polymers-13-00821-f018:**
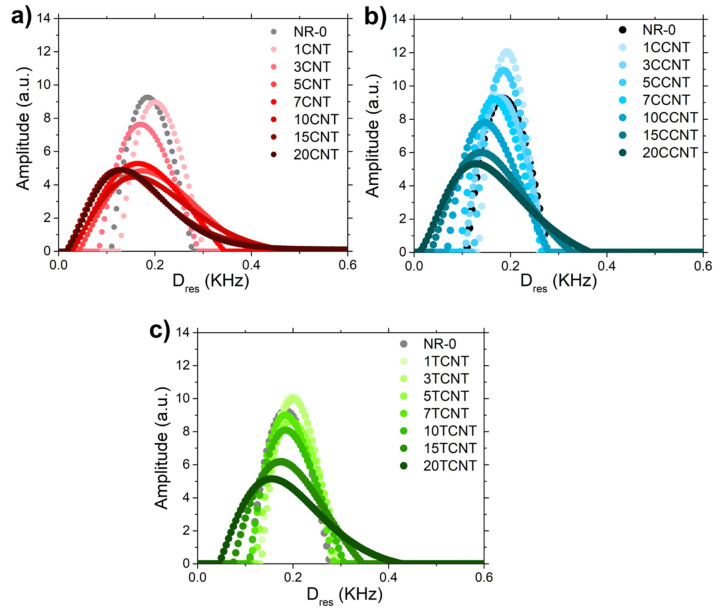
Distributions of the residual dipolar couplings obtained by numerical inversion analysis of *I_nDQ_* via Tikhonov regularization for (**a**) NR-CNT; (**b**) NR-CCNT, and (**c**) NR-TCNT compounds, respectively.

**Figure 19 polymers-13-00821-f019:**
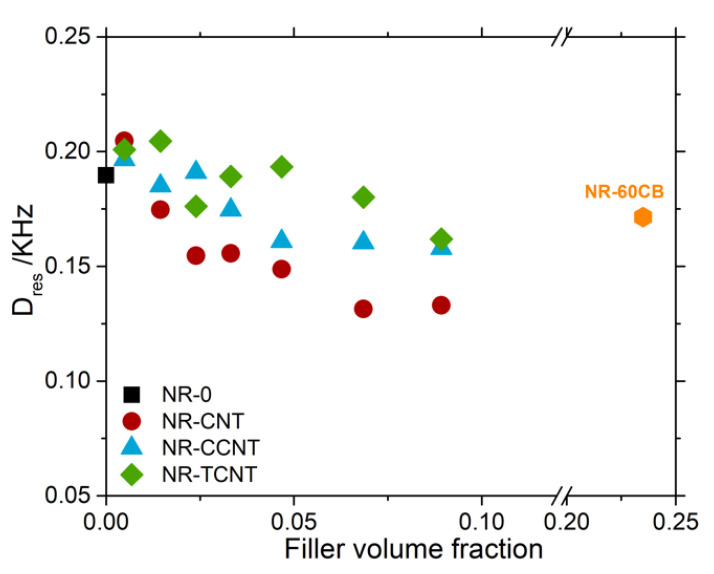
Evolution of residual dipolar coupling value of the distribution maximum as a function of the filler volume fraction of the NR compounds filled with pristine and functionalized CNTs.

**Figure 20 polymers-13-00821-f020:**
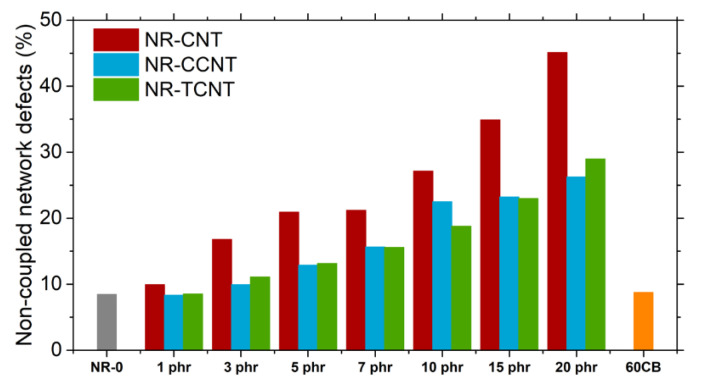
Non-coupled network defects of NR/CNT, NR/CCNT, and NR/TCNT compounds.

**Figure 21 polymers-13-00821-f021:**
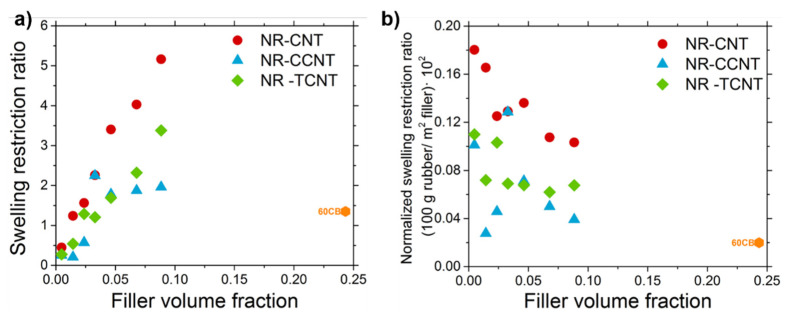
(**a**) Swelling restriction ratio and (**b**) normalized swelling restriction ratio for the studied compounds.

**Table 1 polymers-13-00821-t001:** Formulation of the rubber compounds.

Ingredient	phr			
NR	100	100	100	100
ZnO	3	3	3	3
Stearic Acid	3	3	3	3
CBS	2	2	2	2
S	1	^a^	^a^	1
CNT	1–20	-	-	-
CCNT	-	1–20 ^b^	-	-
TCNT	-	-	1–20 ^b^	-
CB	-	-	-	60

^a^ The amount of sulfur added in the samples filled with modified CNTs varied depending on the quantity of sulfur chemically bonded to the surface of the CNT for each reaction performed. ^b^ The amount of functionalized CNTs added to the rubber recipe was calculated according to the sulfur content to maintain the fraction of CNT constant.

**Table 2 polymers-13-00821-t002:** Fraction of functional groups grafted in the CNT surface according to the S/CNT weight ratio used in the functionalization reaction. Amount of functionalized filler and free sulfur added to the rubber compounds to keep the fraction of CNTs constant.

	CCNTs			TCNTs		
S/CNT Weight Ratio	Grafted Functional Groups (% *w/w*)	f-Filler Added to Rubber Compounds (phr)	Free Sulfur Added to Rubber Compounds (phr)	Grafted Functional Groups (% *w/w*)	f-Filler Added to Rubber Compounds (phr)	Free Sulfur Added to Rubber Compounds (phr)
1:1	29.0	1.41	0.59	25.5	1.34	0.66
1:3	11.3	3.38	0.62	3.8	3.12	0.88
1:5	11.0	5.62	0.38	2.7	5.14	0.86
1:7	9.5	7.73	0.27	4.3	7.32	0.68
1:10	9.5	11.05	---	3.7	10.38	0.62
1:15	10.9	16.83	---	4.5	15.70	0.30
1:20	10.8	22.43	---	4.8	21.01	---

**Table 3 polymers-13-00821-t003:** *I_D_/I_G_* ratios obtained from the Raman spectra of CNTs, CCNTs, and TCNTs.

Sample	I_D_/I_G_ Ratio
CNTs	1.1 ± 0.1
CCNTs	1.2 ± 0.1
TCNTs	1.16 ± 0.1

**Table 4 polymers-13-00821-t004:** Induction and vulcanization time of the studied compounds.

	NR-CNT		NR-CCNT		NR-TCNT	
phr	t_0_	t_97_	t_0_	t_97_	t_0_	t_97_
0	5.92	13.91	5.92	13.91	5.92	13.91
1	4.43	13.01	5.50	11.92	4.76	13.16
3	3.97	10.65	5.33	10.66	4.04	8.37
5	3.36	9.66	3.76	10.61	3.42	8.03
7	2.99	8.6	3.64	10.45	3.15	7.58
10	2.89	8.15	3.32	7.59	2.79	7.43
15	2.55	7.5	2.62	6.83	1.89	7.05
20	2.12	7.32	1.96	6.78	0.37	5.01

## Data Availability

The data presented in this study are available on request from the corresponding author.

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
