# Peer review of "Sulfur-Modified Carbon Nanotubes for the Development of Advanced Elastomeric Materials"

_polymers, 2021, doi:10.3390/polym13050821_

Round 1
Reviewer 1 Report
The manuscript reported a strategy to modify carbon nanotubes with sulfur for preparing advanced rubber composites. This work is interesting. There are, however, several remarks that have to be considered during a revision. These are provided below:
- In Line 74-76, the CCNT and TCNT refer to the chemical oxidation using an acid treatment (Chemically modified CNT) and thermal oxidation of CNT using an oven (Thermal modified CNT), while in the Figure 3 and following discussions, they are used to indicate the sulfur functionalized CNT. It may give some confusion for the readers.
- What is the reaction mechanism between the sulfur and oxi-CNT?How did the author prove that a chemical bond occurred between the two component?
- The TGA curves of the CCNT and TCNT before modifying with sulfur should be provided, because it is difficult to evaluate the weight loss of CCNT or TCNT is from sulfur or the oxidized groups under this circumstance. And the fraction of functional groups grafted in the CNT surface is also controversial.
- The Table 1 is difficult to understand, and why does the grafting rate of CCNT decreases first and then increases as the ratio increases?
- Why is there no S-O peak in Figure 7e?
- In Line 353-357, the author said the CCNT shows severe agglomeration and poor dispersion in the rubber matrix, as observed by SEM images. Why the most remarkable reduction of the Payne effect was found in NR/CCNT samples? These two results seem controversial.
- Besides, there are some grammatical and structural errors. Please proofread and correct.
Author Response
Response to Reviewer 1 Comments
The manuscript reported a strategy to modify carbon nanotubes with sulfur for preparing advanced rubber composites. This work is interesting. There are, however, several remarks that have to be considered during a revision. These are provided below:
1. In Line 74-76, the CCNT and TCNT refer to the chemical oxidation using an acid treatment (Chemically modified CNT) and thermal oxidation of CNT using an oven (Thermal modified CNT), while in the Figure 3 and following discussions, they are used to indicate the sulfur functionalized CNT. It may give some confusion for the readers.
In the revised manuscript the oxidized CNT are named cox-CNT and tox-CNT whereas the sulfur functionalized CNT are named CCNT and TCNT respectively to avoid any possible confusion. This nomenclature was added in the Introduction section (line 74-76) and also in the Experimental section (2.1. Functionalization of CNT).
2.What is the reaction mechanism between the sulfur and oxi-CNT? How did the author prove that a chemical bond occurred between the two component?
The reaction was already reported in the literature (see the references in the manuscript):
- Jin, K.; Zhou, X.; Zhang, L.; Xin, X.; Wang, G.; Liu, Z. Sulfur/carbon nanotube composite film as a flexible cathode for lithium-sulfur batteries. J. Phys. Chem. C 2013, 117, 21112–21119, doi:10.1021/jp406757w.
- Yan, J.; Liu, X.; Wang, X.; Li, B. Long-life, high-efficiency lithium/sulfur batteries from sulfurized carbon nanotube cathodes. J. Mater. Chem. A 2015, 3, 10127–10133, doi:10.1039/c5ta00286a.
For additional information, please revise also the works cited in these two papers, e.g.,
- Ji, L. W.; Rao, M. M.; Zheng, H. M.; Zhang, L.; Li, Y. C.; Duan, W. H.; Guo, J. H.; Cairns, E. J.; Zhang, Y. G. J. Am. Chem. Soc.2011, 133, 18520-18525
- Xin, L. Gu, N. Zhao, Y. Yin, L. Zhou, Y. Guo and L. Wan, J. Am. Chem. Soc. 2012, 134, 18510.
- Fanous, M. Wegner, M. S. Spera and M. R. Buchmeiser, Electrochem. Soc. 2013, 160, A1169.
- Zhang, D. Qiao, J. Pan, Y. Cao, H. Yang and X. Ai, Electrochim. Acta 2013, 87, 497.
- Duan, W. Wang, A Wang, K. Yuan, Z. Yu, H.Zhao, J. Qiu and Y. Yang, J. Mater. Chem. A 2013, 1, 13261.
- Wang, J. Yang, C. Wan, K. Du, J. Xie and N. Xu, Adv. Funct. Mater. 2013, 13, 487.
The attachment of sulfur to the CNT surface after the functionalization reaction and subsequent washing process was proven by observing the presence of C-S and O-S bonds by using X-ray photoelectron spectroscopy (XPS). It was already reported in the original manuscript submitted to Polymers (section 3.1.3. X-ray photoelectron spectroscopy (XPS)).
3.- The TGA curves of the CCNT and TCNT before modifying with sulfur should be provided, because it is difficult to evaluate the weight loss of CCNT or TCNT is from sulfur or the oxidized groups under this circumstance. And the fraction of functional groups grafted in the CNT surface is also controversial.
TGA curves of CNTs after chemica oxidation (cox-CNT) and thermal oxidation treatments (tox-CNT) were added to Figure 5.
4.- The Table 1 is difficult to understand, and why does the grafting rate of CCNT decreases first and then increases as the ratio increases?
The reviewer´s comment refers to Table 2 because an error in the call of tables in the manuscript. This error was addressed in the revised version. This table reports:
- The fraction of grafted functional groups according to the sulfur/CNT weight ratio in the reaction. In all cases, the highest grafted functional group is obtained at the ratio S/CNT 1:1, whereas samples with lower S/CNT ratios only show small differences without a clear trend. It is not possible a deeper discussion about these results.
- The actual amount of functional CNT added to the rubber compounds. The fraction of CNT for each sample should be 1,3 5, 7, 10, 15 and 20 phr. In the case of functionalized fillers, it is important to consider the attached sulfur at the surface in order to maintain the same amount of CNT for all the studied samples.
- The actual amount of free sulfur added to the rubber compounds. The fraction of sulfur for all samples should be 1 phr. In the case of functionalized fillers, it is important to consider the sulfur attached at the CNT surface.
5.- Why is there no S-O peak in Figure 7e?
Thermal oxidation is less effective than chemical oxidation for CNT treatment. For that reason, the amount of grafted functional groups is much lower, reducing the signal-to-noise ratio in the XPS spectra. For that reason, it is more complex to identify and quantify the S-O peak in the TCNT samples.
6.- In Line 353-357, the author said the CCNT shows severe agglomeration and poor dispersion in the rubber matrix, as observed by SEM images. Why the most remarkable reduction of the Payne effect was found in NR/CCNT samples? These two results seem controversial.
We would like to thanks the reviewer for this interesting comment, because it provides one of the key-factor for understand some of the results showed in this work. CCNT are more agglomerated according to the SEM images, something that is also related with the conductivity results.
On the other hand, the Payne effect shows the decay of the elastic modulus of rubber compounds with the strain sweep because of the breakdown of filler aggregates. This effect is strongly reduced in the case of CCNT.
These combined results reflect that CCNT would be agglomerated during the modification reaction, creating strong interactions between CNT particles (maybe covalent). Consequently, the CNT agglomerates are bigger than pristine and TCNT (for that reason the SEM images, lower conductivity at low CNT fractions and lower mechanical properties) but they cannot be easily broken during the strain sweep (because of the strong interactions caused during the modification reaction), reducing the so-name Payne effect.
7.- Besides, there are some grammatical and structural errors. Please proofread and correct
English grammar was revised. All changes were highlighted in the revised manuscript.

Reviewer 2 Report
Dear Authors,
Your work is very interesting and well prepared. I have only few comments and suggestions:
- Lines: 155, 269, 346, 374, 419, 451, 480, 590, 609, 628, 638, 664, 676- Reference missing
- The induction time (t0) and vulcanization time (t97) for all prepared materials should be presented in table. In my opinion the scorch time (ts2), optimum vulcanization time (t90), curing rate index (CRI) and percentage of reversion after 300s form optimum curing time (R300) - it permit to obtain valuable informations about the processing parameters
- How long does it take to introduce the CNTs into rubber mixtures (especially when their amount were higher than 10 phr)? The filler was added after or befor sulfur? Did you use any other plasticizer than stearic acid?
- Crosslink density using Flory-Rehner relationship (by equilibrium swelling) should be determined and discussed with the results of DMA. Some informations about crosslink density can be obtained by the analysis of the difference between maximum and minimum torque (from rheometric measurements of rubber mixtures)
- What are advanateges and disadvantages of applied by you functionalization of CNTs in comparison to other chemical (or physical) modifications of CNT for natural rubber mixtures? (In the context of processing or physical properties)
Author Response
Response to Reviewer 2 Comments
Dear Authors,
Your work is very interesting and well prepared. I have only few comments and suggestions:
1.- Lines: 155, 269, 346, 374, 419, 451, 480, 590, 609, 628, 638, 664, 676- Reference missing
These errors were addressed in the revised manuscript.
2.- The induction time (t0) and vulcanization time (t97) for all prepared materials should be presented in table. In my opinion the scorch time (ts2), optimum vulcanization time (t90), curing rate index (CRI) and percentage of reversion after 300s form optimum curing time (R300) - it permit to obtain valuable informations about the processing parameters.
Table with the induction and vulcanization time was added in the revised manuscript.
Thank you for the suggestion of the use of the scorch time, vulcanization time, CRI and R300 , to obtain valuable information about the processing of the compounds. We will take it into account to obtain an optimum analysis of the vulcanization reactions for future works.
3.- How long does it take to introduce the CNTs into rubber mixtures (especially when their amount were higher than 10 phr)? The filler was added after or befor sulfur? Did you use any other plasticizer than stearic acid?
Rubber compounds were prepared on a laboratory two-roll mill with friction ratio of 1:1.15. In all cases the compounding takes around 15 minutes that is the time required to obtain optimum filler dispersion for those samples with the highest filler content. Fillers always were added before the addition of sulfur , that is the last ingredient to be added during the rubber compounding in order to avoid prevulcanization processes. In the case of functionalized CNT, the presence of sulfur at the CNT surface may promote prevulcanization reactions, for that reason, the rolls were kept cold during the mixing procedure by circulating cold water (at 20 °C). No other plasticizer than stearic acid was used in the rubber compounds.
4.- Crosslink density using Flory-Rehner relationship (by equilibrium swelling) should be determined and discussed with the results of DMA. Some informations about crosslink density can be obtained by the analysis of the difference between maximum and minimum torque (from rheometric measurements of rubber mixtures).
We are sorry to say that reviewer´s criticism about cross-link density has not sense to our best understanding. In this work we have used DQ-NMR experiments to determine the cross-link density of filled rubber compounds. Nowadays, it is probably the most accurate experimental approach to quantify the network structure in filled compounds in terms of: non-elastic network defects (dangling chain ends and loops), cross-link density and spatial distribution of cross-links. DQ-NMR experiments are able to overcome most of the uncertainties that would invalid the application of equilibrium swelling experiments or mechanical analysis for determining the cross-link density in filled rubber samples, because in they are completely dominated by the filler effect and cannot be applied to quantify the cross-link density of rubber matrix (something that is possible in unfilled compounds, although with some uncertainties related with the swelling thermodynamics and the elastic model applied for define the rubber elasticity in swollen samples). We would suggest to the reviewer to revise the huge amount of literature about this issue to clarify this important and basic concept in rubber science and technology.
5.- What are advanateges and disadvantages of applied by you functionalization of CNTs in comparison to other chemical (or physical) modifications of CNT for natural rubber mixtures? (In the context of processing or physical properties)
The main advantage of the functionalization used in this manuscript in comparison to others is that the sulfur-functionalization avoids the introduction of new ingredients in the formulation of the rubber compounds, improving the efficiency of the vulcanization process with pristine CNT and forming covalent bonds with the rubber matrix. As it was observed in our previous works [1,2], rubber compounds filled with pristine CNTs do not achieve the expected performance in terms of rolling resistance because CNTs-rubber interactions are based on adsorptive phenomena, they interfere with the vulcanization chemistry by the absorption of the sulfur in their surface, and also because the trend of CNTs to aggregate. Therefore the chemical bonding of the sulfur to the surface of the particles, avoids two of these specific problems observed for CNTs.
- Bernal-Ortega, P.; Bernal, M.M.; González-Jiménez, A.; Posadas, P.; Navarro, R.; Valentín, J.L. New insight into structure-property relationships of natural rubber and styrene-butadiene rubber nanocomposites filled with MWCNT. Polymer (Guildf). 2020, 201, doi:10.1016/j.polymer.2020.122604.
- Bernal-Ortega, P.; Bernal, M.M.; González-Jiménez, A.; Posadas, P.; Navarro, R.; Valentín, J.L. Erratum to “New insight into structure-property relationships of natural rubber and styrene-butadiene rubber nanocomposites filled with MWCNT” [Polymer 201 (2020) 122604] (Polymer (2020) 201, (S0032386120304353), (10.1016/j.polymer.2020.122604)). Polymer (Guildf). 2020, 203, doi:10.1016/j.polymer.2020.122720.

Reviewer 3 Report
The authors have done a thorough study on their work entitled “Sulfur-modified carbon nanotubes for the development of advanced elastomeric materials better dispersion.” The authors intended to obtain better rubber/filler interactions by modified CNT. However, the mechanical properties, electrical conductivity (percolation threshold), reinforcing degree, etc., were not improved with the modification compared with the neat CNT-filled case, despite the rolling resistance was found to be reduced after modification. Thus, there is a need to clearly clarify the purpose and finding.
Author Response
Response to Reviewer 3 Comments
The authors have done a thorough study on their work entitled “Sulfur-modified carbon nanotubes for the development of advanced elastomeric materials better dispersion.” The authors intended to obtain better rubber/filler interactions by modified CNT. However, the mechanical properties, electrical conductivity (percolation threshold), reinforcing degree, etc., were not improved with the modification compared with the neat CNT-filled case, despite the rolling resistance was found to be reduced after modification. Thus, there is a need to clearly clarify the purpose and finding.
The purpose is clearly addressed in the first lines of the abstract “The outstanding properties of carbon nanotubes (CNT) present some limitations when introduced into rubber matrices, especially when these nano-particles are applied in high-performance tire tread compounds. Their tendency to agglomerate into bundles due to van der Waals interactions, the strong influence of CNT on the vulcanization process and the adsorptive nature of filler-rubber interactions contribute to increase the energy dissipation phenomena on rubber-CNT compounds. Consequently, their expected performance in terms of rolling resistance is limited. To overcome these three important issues, the CNT have been surface-modified with oxygen-bearing groups and sulfur, resulting in an improvement of the key properties of these rubber compounds for their use in tire tread applications.”
The finding is also well understood by the reviewer, the use of functionalized CNT provides a better performance of rubber compounds in terms of rolling resistance because the energy dissipation phenomena are minimized. The latter should be related to two main reasons: i) the effect of pristine CNT on the vulcanization process is drastically reduced (as it was reported in terms of the cross-link density and non-elastic network defects, i.e. dangling chain ends) and ii) it is possible to create covalent interactions between the rubber chains and the CNT surface. The latter minimizes the energy dissipation phenomena cause by adsorption-desorption mechanisms that dominates the interactions in the rubber compounds filled with pristine CNT.
These statements were pointed out in the conclusions “The aim of this study was to obtain a better interaction of CNT with the rubber matrix by the formation of covalent bonds through the surface modification of these particles. The sulfur modification of the CNT is a promising functionalization method that can be scala-ble to an industrial process. It was observed that the use of sulfur based CNT is a suitable approach that partially overcomes two of the problems identified in the rubber com-pounds filled with pristine CNT that may limit their use in high performance tire tread compounds. The use of functionalized CNT with sulfur reduces the adsorption of acceler-ants and promotes the formation of covalent bonds between the rubber chains and the CNT surface during the vulcanization process. In this way, these modified nanoparticles not only act like fillers but also as macro-crosslinks that are able to create a rubber network with higher crosslink density and lower fraction of non-elastic network defects as com-pared with the rubber compounds filled with pristine CNT. As a consequence, these functionalized CNT lead to an improvement in the viscoelastic properties of rubber com-pounds by reducing their energy dissipation and promoting a substantial decrease in the loss factor at 60 ºC, that is a key parameter for the rolling resistance performance of tire tread rubber compounds.”.
Finally, it is important to say that sulfur functionalization of CNT is not able to solve the trend of pristine CNT to aggregates. For that reason, it is not possible to improve the mechanical properties and electrical conductivity of pristine CNT (which are exceptionally improved related to traditional fillers such as carbon black). It was also stated in the conclusions “On the opposite side, the functionalization with sulfur is not able to solve the trend of pristine CNT to be agglomerate into bundles, causing a critical reduction of the effective shape factor with the filler volume fraction that have important consequences in their rein-forcing properties. Nevertheless, the modified nanoparticles seem to form covalent bonds between them, increase the resistance of CNT aggregates to be broken during rubber de-formation and reducing the energy dissipation phenomena attributed to the breakdown of filler interactions.”

Reviewer 4 Report
Bernal-Ortega et al. reported the manuscript entitled ‘‘Sulfur-modified carbon nanotubes for the development of advanced elastomeric materials.’’ This manuscript needs major revision before publication. Some comments are as follow:
- Abstract should be more quantitative.
- Please mention the novelty of the current study.
- The introduction part should be improved.
- If possible the authors need to provide lower magnification SEM images to check the homogeneous dispersion of CNT in the fabricated samples.
- In line 126, Change from 2h to 2 h maintain consistency in the whole manuscript.
- Please explain ‘’Error! Reference source not found’’…
- Why CCNT showed lower thermal stability compared with TCNT, please explain?
- In line 293 author used aggressive oxidation…please rewrite this statement.
- After surface modification of CNT (thermally and chemically) still aggregation was observed and showed lower thermal stability. Please explain the advantages of modification clearly.
Author Response
Response to Reviewer 4 Comments
Bernal-Ortega et al. reported the manuscript entitled ‘‘Sulfur-modified carbon nanotubes for the development of advanced elastomeric materials.’’ This manuscript needs major revision before publication. Some comments are as follow:
1.- Abstract should be more quantitative.
This comment seems to suggest the incorporation to the abstract to some quantification of the enhancement provoked by the modification of CNT according to the sentence “The outcome of this research revealed that the formation of covalent bonds between the rubber matrix and the nano-particles by the introduction of sulfur at the CNT surface has positive effects on the viscoelastic behavior and the network structure of the rubber compounds, by a decrease of both the loss factor at 60 ËšC (rolling resistance) and the non-elastic defects, while increasing the crosslink density of the new compounds.”. Nevertheless, the decrease of loss factor and the non-elastic defects, and the increasing of crosslink density with the use of functionalized CNT shows a clear dependence with the CNT volume fraction and the modification method. For that reason, it was not included in the abstract, although these values are easily obtained in the reported results.
2.- Please mention the novelty of the current study.
It was done in the introduction section
3.- The introduction part should be improved.
The introduction section was modified in order to clarify the novelty of this study.
4.- If possible the authors need to provide lower magnification SEM images to check the homogeneous dispersion of CNT in the fabricated samples.
With lower magnification SEM images, it is not possible to see the CNTs aggregates, observing only the rubber surface. For that reason, these images are not added to the revised manuscript. The quality for the CNT dispersion could be also evaluated by using electrical conductivity (as it was done in the manuscript).
5.- In line 126, Change from 2h to 2 h maintain consistency in the whole manuscript.
The change was made to maintain the consistency in the manuscript.
6.- Please explain ‘’Error! Reference source not found’’…
The references were updated and the “Error …” problem was solved.
7.- Why CCNT showed lower thermal stability compared with TCNT, please explain?
The chemical oxidation of the particles, with the use of strong acids, causes a higher damage in the sidewalls of the carbon nanotubes reducing their thermal stability in comparison with the weaker thermal oxidation process.
8.- In line 293 author used aggressive oxidation…please rewrite this statement.
The statement was modified in the line 293.
9.- After surface modification of CNT (thermally and chemically) still aggregation was observed and showed lower thermal stability. Please explain the advantages of modification clearly.
The main advantages of the modification reside in the formation of covalent bonds between the rubber chains and the filler particles and the more efficient use of sulfur during the vulcanization process. 1H DQ-NMR experiments showed how NR samples filled with TCNT present higher values of crosslink density and less dangling chain ends than the compounds filled with pristine CNT. These non-elastic network defects in addition to rubber bonding-debonding mechanisms at the CNT surface (caused by the adsorptive nature of their interactions) act as energy dissipation elements. In consequence, the overcome of these limitations by using functionalized CNT with elemental sulfur is able to enhance the rolling resistance performance of these compounds as can be observed by the decrease of the loss factor at 60 °C compared to rubber compounds filled with pristine CNT.
These statements are clearly described in the revised manuscript.

Round 2
Reviewer 1 Report
This manuscript has been carefully revised. However, there is still a problem and concern about the SEM image and Payne effect. The author said that "CCNT would be agglomerated during the modification reaction, creating strong interactions between CNT particles. Consequently, the CNT agglomerates are bigger than pristine and TCNT but they cannot be easily broken during the strain sweep (because of the strong interactions caused during the modification reaction), reducing the so-name Payne effect." But why the G' of CCNT composites is lower than CNT at a very small strain? In fact, as long as the composites contain more agglomerates, the pristine G' will be higher, but this is not the case.
Author Response
This manuscript has been carefully revised. However, there is still a problem and concern about the SEM image and Payne effect. The author said that "CCNT would be agglomerated during the modification reaction, creating strong interactions between CNT particles. Consequently, the CNT agglomerates are bigger than pristine and TCNT but they cannot be easily broken during the strain sweep (because of the strong interactions caused during the modification reaction), reducing the so-name Payne effect." But why the G' of CCNT composites is lower than CNT at a very small strain? In fact, as long as the composites contain more agglomerates, the pristine G' will be higher, but this is not the case.
In page 16 it is written the answer to the reviewer question: “The combination of these results seems to indicate that during the surface modification reaction of CNT, these nanoparticles may tend to agglomerate and also to create strong chemical bonds between them. As consequence, the primary particles of f-CNT that are added to the rubber matrix during the compounding step may be in fact composed by a group of inseparable nanotubes that has been chemically linked during the modification reaction, increasing the size and reducing the effective shape factor of the pristine CNT.”
We would like to provide a simplified example (oversized the dimensions and consequences respect to the CCNT case) in order to try to clarify this statement to the reviewer. Carbon black has bigger size than CNT. If these fillers are used as fillers in rubber compounds (by adding the same volume fraction), the sample filled with carbon black will show lower G’ at the minimum deformation than CNT. It is because the filler network (filler-filler interactions) is formed at lower percolation threshold for fillers with smaller particle size.
In the current work, CCNT has bigger particle size than pristine CNT. The aggregation and formation of covalent bonds between the nanoparticles during the functionalization reaction makes possible to create unbreakable bigger particles that is afterword added to the rubber matrix. For that reason, rubber compounds with CCNT as filler shows lower G’ at the minimum deformation as compared to CNT-rubber compounds with the same filler volume fraction.
Reviewer 4 Report
The authors have addressed all the major concerns, so I recommended it for publication.
Author Response
Authors would like to acknowledge the reviewer's comments and suggestions. They have improved the quality of the paper.